# Bayesian learning of chemisorption for bridging the complexity of electronic descriptors

Siwen Wang[1,2], Hemanth Somarajan Pillai[1,2] & Hongliang Xin [1✉]

Building upon the *d*-band reactivity theory in surface chemistry and catalysis, we develop a Bayesian learning approach to probing chemisorption processes at atomically tailored metal sites. With representative species, e.g., *O and *OH, Bayesian models trained with ab initio adsorption properties of transition metals predict site reactivity at a diverse range of intermetallics and near-surface alloys while naturally providing uncertainty quantification from posterior sampling. More importantly, this conceptual framework sheds light on the orbitalwise nature of chemical bonding at adsorption sites with *d*-states characteristics ranging from bulk-like semi-elliptic bands to free-atom-like discrete energy levels, bridging the complexity of electronic descriptors for the prediction of novel catalytic materials.

[1] Department of Chemical Engineering, Virginia Polytechnic Institute and State University, Blacksburg, VA 24061, USA. [2]These authors contributed equally: Siwen Wang, Hemanth Somarajan Pillai. ✉email: hxin@vt.edu

Adsorption of molecules or their fragments at transition-metal surfaces is a fundamental process for many technological applications, such as chemical sensing, molecular self-assembly, and heterogeneous catalysis. Because of the convoluted interplay between electron transfer and orbital coupling, chemical bonding can be formidably complex. Recent decades have brought major advances in spectroscopic tools[1,2], which reveal orbitalwise information of chemisorbed systems and concurrently in predicting chemical reactivity at sites of interest via electronic factors, e.g., the number of valence $d$-electrons[3], density of $d$-states at the Fermi level[4], $d$-band center[5], and $d$-band upper edge[6,7]. Compared with a full quantum-mechanics treatment of many-body systems, the simplicity of physics-inspired descriptors comes at a cost of limited generalization, particularly for high-throughput materials screening. Incorporation of multifidelity site features into reactivity models with machine learning (ML) algorithms has shown early promise for the prediction of adsorption energies, with an accuracy comparable to the typical error (~0.1−0.2 eV) of density functional theory (DFT) calculations[8–16]. However, the approach is largely black-box in nature, prohibiting its physical interpretation. Developing a theory-based, generalizable model of chemisorption that bridges the complexity of electronic descriptors, and predicts the binding affinity of active sites to key reaction intermediates with uncertainty quantification represents one of the biggest challenges in fundamental catalysis.

Here, we present a Bayesian inference approach to probing chemisorption processes at metal sites by learning from ab initio datasets. The model is built upon the basic framework of the $d$-band reactivity theory[5], while employing a Newns–Anderson-type Hamiltonian[17,18] to capture essential physics of adsorbate-substrate interactions. Such types of simplified Hamiltonians were originally used for describing magnetic properties of impurities in a bulk metallic host[17], and later extended with success by Newns and Grimley to chemisorption at surfaces[18,19]. A basis set of orbitals consisting of the adsorbate and substrate states was used for solving the hybridization problem within a self-consistent Hartree–Fock scheme[18]. Despite a remarkable success in advancing the basic understanding of adsorption phenomena at surfaces, particularly for $d$-block metals[6], its application in materials design remains limited due to the lack of accurate model parameters and meaningful error estimates. Bayesian inference produces the posterior probability distribution of model parameters under the influence of observations and prior knowledge[20]. With representative species, e.g., *O and *OH, we demonstrate the predictive performance and physical interpretability of Bayesian models for chemical bonding at a diverse range of intermetallics and near-surface alloys, bridging the complexity of electronic descriptors in search of novel catalytic materials.

## Results

### The $d$-band reactivity theory.

Within the basic framework of the $d$-band reactivity theory for transition-metal surfaces, the formation of the adsorbate-metal bond conceptually takes place in two consecutive steps[5], as illustrated in Fig. 1. First, the adsorbate frontier orbital (or orbitals) $|a\rangle$ at $\epsilon_a^0$ couples to the delocalized, free-electron-like $sp$-states of the metal substrate, leading to a Lorentzian-shaped resonance state at $\epsilon_a$. Second, the adsorbate resonance state interacts with the localized, narrowly-distributed metal $d$-states, shifting up in energies due to the orthogonalization penalty for satisfying the Pauli principle, and then splitting into bonding and antibonding states. The first step interaction contributes a constant $\Delta E_0$ albeit often the largest part of chemical bonding. The variation in adsorption energies from one metal to another is determined by the metal $d$-states. This part of

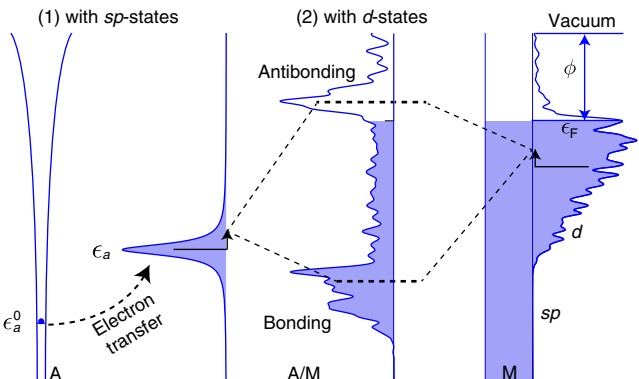

**Fig. 1 Illustration of chemical bonding at transition-metal surfaces within the $d$-band reactivity theory.** An adsorbate A with a valence electron at a discrete energy level $\epsilon_a^0$ first interacts with the free-electron-like $sp$-states of the substrate M, forming a broadened resonance at $\epsilon_a$ accompanied with electron transfer. Conceptually, it further overlaps and hybridizes with the narrowly distributed $d$-states, which leads to a splitting into bonding and antibonding states. The work function $\phi$ and Fermi level $\epsilon_F$ of M are marked.

the interaction energy $\Delta E_d$ can be further partitioned into orbital orthogonalization and orbital hybridization contributions[21]. To a first approximation, the orbital hybridization energy can be evaluated by the changes of integrated one-electron energies. The orbital orthogonalization cost is considered simply as proportional to the product of interatomic coupling matrix and overlap matrix, $VS$, or equivalently $\alpha V^2$, where $\alpha$ is the orbital overlap coefficient. The absolute value of $V^2$ can be written as $\beta V_{ad}^2$, in which the standard values of $V_{ad}^2$ relative to Cu are readily available on the Solid State Table[22]. The overall adsorption energy $\Delta E$ can then be written as the sum of the energy contributions from the $sp$-states $\Delta E_0$ and the $d$-states $\Delta E_d$, with the latter depending on the symmetry and degeneracy of adsorbate frontier orbitals. Another important information from this framework is the evolving density of states projected onto the adsorbate orbital (s) upon adsorption, $\rho_a$. A full account of the theoretical framework is presented in the "Methods" section.

There are a number of unknown parameters within the basic framework of the $d$-band reactivity theory as discussed above and detailed in "Methods" section, including the energy contribution from the $sp$-band $\Delta E_0$, adsorbate resonance energy $\epsilon_a$ relative to the Fermi level, $sp$-band chemisorption function $\Delta_0$, orbital overlap coefficient $\alpha$, and orbital coupling coefficient $\beta$. By least-squares fitting of the adsorbate density of states and the integrated one-electron energy changes to those from DFT calculations[23,24], the Schmickler model of electron transfer has been developed to understand $H_2$ evolution/oxidation and $OH^-$ adsorption at metal–electrolyte interfaces. However, the deterministic fitting of adsorption properties from a single surface is prone to overfitting or trapping into a locally optimal region, limiting its application in catalysis.

### Bayesian learning.

We instead employ Bayesian learning to infer the vector of model parameters $\vec{\theta} = (\Delta E_0, \epsilon_a, \Delta_0, \alpha, \beta)'$ from the evidence, i.e., ab initio adsorption properties, along with prior knowledge if available[20]. In Bayes' view, those parameters are not deterministic point values, but rather probabilistic distributions reflecting the uncertainty of physical variables. The use of parameter distributions as opposed to computationally-derived point values has obvious advantages for uncertainty quantification. In the chemical sciences, Bayesian learning has been used for calibration and validation of thermodynamic models for the uptake

of $CO_2$ in mesoporous silica-supported amines[25], designing the Bayesian error estimation functional with van der Waals correlations[26], and identifying potentially active sites and mechanisms of catalytic reactions[27], just to name a few. The Bayesian approach allows one to infer the posterior probability distribution $P(\vec{\theta}|\mathcal{D})$ for latent variables based on the prior $P(\vec{\theta})$ as well as the likelihood function $P(\mathcal{D}|\vec{\theta})$ subject to the observation $\mathcal{D}$. The mathematical relationship between the prior, observation, and posterior is given by the Bayes' theorem[20], $P(\vec{\theta}|\mathcal{D}) = P(\mathcal{D}|\vec{\theta})P(\vec{\theta})/P(\mathcal{D})$. Our initial belief about likely parameter values is provided by weakly informative priors to minimize potential bias. For example, $\Delta E_0$ and $\epsilon_a$ can be estimated from DFT calculations of the adsorbate on a simple metal, e.g., sodium (Na) at the face-centered cubic (fcc) phase. Specifically, we took Normal for floating-point variables unrestricted in sign, Log-Normal for non-negative parameters, and Uniform for others (see the details of Bayesian learning and parameter choices in the "Methods" section). Computing the normalizing constant $P(\mathcal{D})$, denominator of the posterior distribution, is impossible in most practical scenarios. To avoid this complication, the Markov chain Monte Carlo (MCMC) method[28], whose sampling criterion only depends on the relative posterior density of the newly explored point and its preceding point, is used. To compute the transition probability of each MCMC step, we define the sum of the (negative) logarithm of the likelihood functions corresponding to binding energies and projected density of states onto each adsorbate orbital with a hyperparameter $\lambda$ adjusting the weight of two contributing metrics, see details in the "Methods" section. After a large number of MCMC samplings, burning (discard) of the first half of the trajectory and then thinning (1 out of 5 samplings) were performed before extracting converged values from the joint posterior distributions. The convergence of the MCMC sampling is checked by using parallel chains with different starting parameter sets such that the variance of interchain samplings is close or within 1.2–1.5 times to that of intrachains[28]. The complete code, named *Bayeschem*, is now available at a Github repository https://github.com/hlxin/bayeschem for public access.

**Model development**. In Fig. 2a, we are showing the co-variance of the joint posterior distribution for each parameter pair and the 1D histogram of model parameters ($\Delta E_0$, $\epsilon_a$, $\Delta_0$, $\alpha$, and $\beta$) from MCMC simulations for *O adsorption at the fcc-hollow site of the {111}-terminated transition-metal surfaces (Cu, Ag, Au, Ni, Pd, Pt, Co, Rh, Ir, and Ru). We assume three degenerate $O_{2p}$ orbitals as used before[29] for demonstration of the approach, while later extend it to multiorbital models. To attain converged posterior distributions, 200k MCMC sampling steps with the Metropolis–Hastings algorithm were performed in a multi-dimensional parameter space illustrated in Fig. 2b. In Fig. 2, the approximate contours for 68, 95, and 99% confidence regions are shown at the lower triangle, showing little to no correlation between latent-variable pairs.

With the converged Bayesian sampling, in Fig. 3a, it shows the model-predicted adsorption energies of *O at the fcc-hollow site of transition-metal surfaces, with a mean absolute error (MAE) ~0.17 eV compared to DFT calculations. The standard deviation of model prediction using the posterior distribution of model parameters ($\vec{\theta}$, $\vec{\sigma}$) is overlaid, providing for the first time uncertainty quantification of adsorption energies within the $d$-band reactivity theory. Figure 3b shows DFT-calculated and model-constructed projected density of states onto the $O_{2p}$ orbital using the posterior means of model parameters, taking Pt(111) as an example (see all the surfaces in Supplementary Fig. 1). The

chemisorption function $\Delta(\epsilon)$ and its Hilbert transform $\Lambda(\epsilon)$ along with the straight adsorbate line ($\epsilon - \epsilon_a$) are shown for the graphical solution of the Newns–Anderson model[18]. The intersects indicated by solid circles in Fig. 3b represent the $O_{2p}$–$Pt_{5d}$ bonding and antibonding states, with the latter above the Fermi level, suggesting a strong covalent interaction of *O at Pt(111). Given the simplicity of the model, the clearly captured electronic structure of the adsorbate–substrate system and the reactivity trend are satisfying.

To extend the approach for adsorbates with multiple valence orbitals that possibly contribute to bonding, we have explicitly treated $O_{2p}$ states with the doubly degenerate $p_{xy}$ orbitals and the single $p_z$ orbital in Bayesian learning. We infer model parameters ($\epsilon_a$, $\Delta_0$, and $\beta$) corresponding to each non-equivalent adsorbate orbital together with an orbital-independent $\alpha$[29] and a global parameter $\Delta E_0$. The posterior parameter distributions are shown in Supplementary Fig. 2. From the posterior means of model parameters, we can see that the orbital coupling coefficient $\beta$ of $p_{xy}$ (1.67 $eV^{-1}$) is smaller than that of $p_z$ (1.77 $eV^{-1}$), consistent with the symmetry analysis, that the $p_{xy}$ orbitals that are parallel to a surface form $\pi$ bonds with the $d$-states, while the $p_z$ orbital can interact through a stronger $\sigma$ bond. A weaker coupling manifests itself in a narrower orbital splitting of $\pi/\pi^*$ than that of $\sigma/\sigma^*$, which has been previously observed using the angle-resolved photoemission spectroscopy on Cu and Ni[30]. In Supplementary Figs. 3 and 4, it shows that the model-constructed projected density of states onto symmetry-resolved orbitals closely resemble the DFT-calculated distributions and the predicted values of *O adsorption energies have a MAE ~0.17 eV. To demonstrate the robustness and generalizability of the approach, we have also optimized the *Bayeschem* model of *O at the atop configuration, see Supplementary Figs. 5–7. In this model scheme, an individual set of parameters is obtained for the adsorbate at a given site. Compared to the linear adsorption-energy scaling relations[31] that link adsorption energies of different adsorbates, *Bayeschem* creates the connection between the electronic structure of a surface site and the adsorption energy.

To test the prediction capability of the *Bayeschem* model for unseen systems, we took the *OH species at the atop adsorption configuration as a case study because of its fundamental importance in understanding the nature of chemical bonding[32], and practical interests as a key reactivity descriptor in transition metal catalysis[33–35]. Three frontier molecular orbitals, i.e., $3\sigma$, $1\pi$, and $4\sigma^*$, are assumed to be involved in chemical bonding[32]. Symmetry-resolved, molecular orbital density of states projected onto OH along with adsorption energies are used as the DFT ground truth $Y$ in Eq. (6). With the *Bayeschem* model developed here (see Supplementary Figs. 8–10 for posterior parameter distributions, model-predicted adsorption energies and projected density of states on training samples), we predict *OH binding energies at a diverse range of intermetallics and near-surface alloys. Specifically, we included $A_3B$, $A'@A_{ML}$, $A$-$B@A_{ML}$, $A_3B@A_{ML}$, $A@A_3B$, and $A@AB_3$, where A (A′) represents ten fcc/hcp metals used in the model development and B covers $d$-metals across the periodic table (see ref. 36 for structural details and tabulated data). The coupling matrix element $V_{ad}$ for alloys is assumed to be constant from the Solid State Table[22]. Its dependence on the local chemical environment can be incorporated into the model using the tight-binding approximation[33]. The A sites of above-mentioned surfaces exhibit diverse characteristics of the metal $d$-states ranging from bulk-like semi-elliptic bands to free-atom-like discrete energy levels[37], as illustrated in Fig. 4a using Pt and $Ag_3Pt$ as examples. Similar to previous observations of single-atom alloys with coinage metal hosts[37,38], a reactive guest metal often exhibits peaky signatures within the $d$-band due to the energy misalignment of coupling $d$–

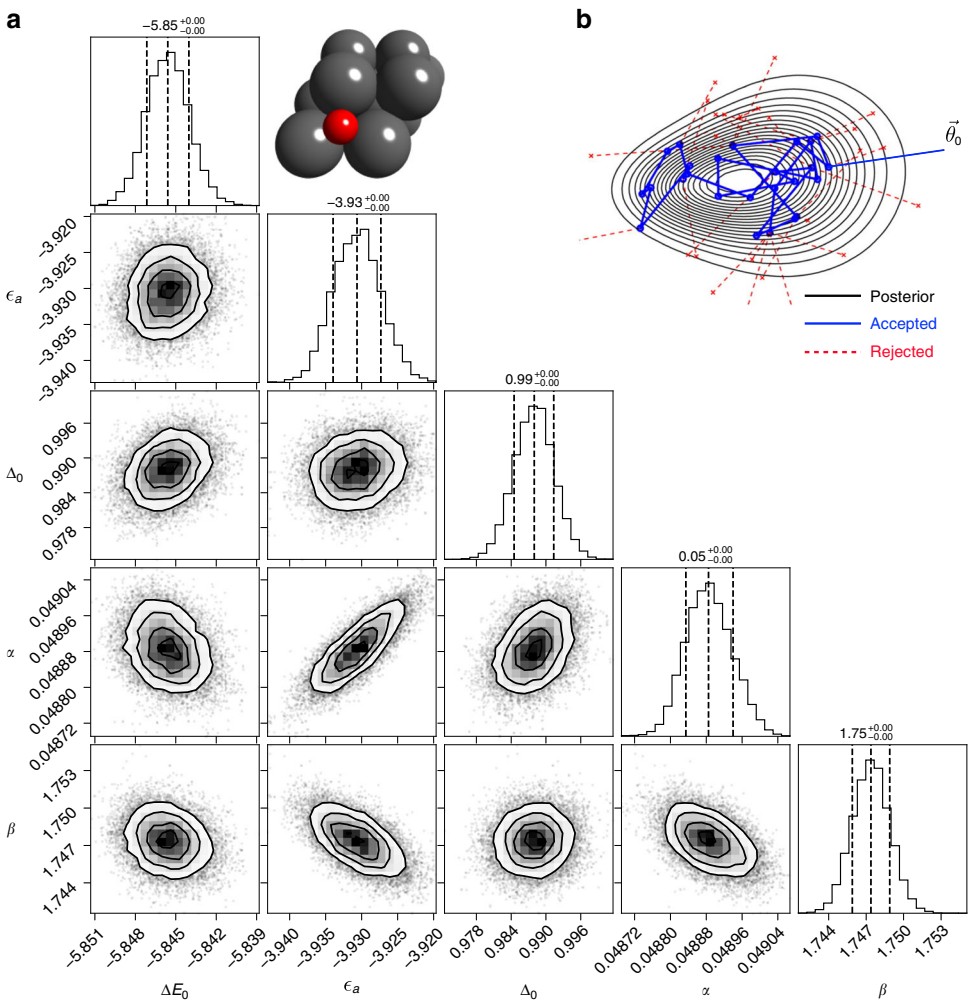

**Fig. 2 Bayesian parameterization. a** The co-variance of the joint posterior distribution for each parameter pair and the 1D histogram of model parameters ($\Delta E_0$, $\epsilon_a$, $\Delta_0$, $\alpha$, and $\beta$) from MCMC simulations for *O adsorption at the fcc-hollow site of the {111}-terminated transition-metal surfaces. A top view of the model structure is shown in inset. **b** Schematic illustration of the MCMC sampling in a multidimensional parameter space. $\vec{\theta}_0$ represents the initial guess of model parameters.

$d$ orbitals[7]. A direct consequence of such diverse electronic properties of adsorption sites is that no single electronic descriptor can capture the local chemical reactivity accurately. Encouragingly, the *Bayeschem* model, parameterized using ten pristine transition-metal data, predicts *OH adsorption energies on 512 alloy surfaces with a MAE 0.16 eV, see Fig. 4b. The standard deviation of predicted *OH adsorption energies from the posterior distribution of model parameters is marked for uncertainty quantification. It shows a similar performance to data-driven ML models[8–11] while outperforming the state-of-the-art electronic descriptors, e.g., the $d$-band center $\epsilon_d$ (MAE: 0.20 eV) and upper edge $\epsilon_u$ (MAE: 0.23 eV). The approach can be easily extended to more complex adsorbates than *O and *OH, e.g., *OOH, without losing its generalizability in the development workflow.

**Orbitalwise interpretation of chemical bonding**. More importantly, the Bayesian framework with built-in physics allows us to quantitatively interrogate the underlying mechanism of chemical bonding, that is difficult to obtain from purely data-driven regression models. Taking *OH adsorption at the M (10 fcc/hcp metals) site of {111}-terminated Ag$_3$M intermetallics as examples, Fig. 4c shows the partition of *OH adsorption energies resulting from the 2$^{nd}$ step interaction ($\Delta E_d$) into orbital orthogonalization

and hybridization. As we can see, for 3$d$, 4$d$, and 5$d$ series of the guest metal M, the orthogonalization and hybridization contributions decrease in magnitude from left to right across the periodic table, while the hybridization dominates the reactivity trends. The changes in $\Delta E_d^{hyb}$ can be understood from the simplified $d$-band model, with the position and occupancy of adsorbate–substrate antibonding states tracking with the $d$-band center or upper edge. The orthogonalization energy is proportional to the filling $f$ and $V_{ad}^2$ (see Eq. (4)), which are offsetting each other to a certain extent ($V_{ad}^2$ decreases while $f$ increases across 3$d$, 4$d$, and 5$d$ series), leading to a less dominant role than the hybridization. The orbitalwise contributions shown in Fig. 4c with different fill patterns suggest that the sole contribution of *OH adsorption at $d$-metal surfaces is from the 1$\pi$ orbital, while those from 3$\sigma$ and 4$\sigma^*$ are too small to be visible. This is supported by projected molecular orbital density of states in Supplementary Fig. 7, which shows that 3$\sigma$ and 4$\sigma^*$ are forming resonance states after their interactions with the $sp$-states of the metal site without noticeable splitting due to $d$-states. Thus, they do not contribute to the observed trend of *OH adsorption. The Bayesian-optimized orbital coupling coefficients of 3$\sigma$ and 4$\sigma^*$ are rather small (0.12 and 0.001 as shown in Supplementary Fig. 5, respectively), supporting unfavorable orbital overlaps with the $d$-states. This rationalizes the observation that

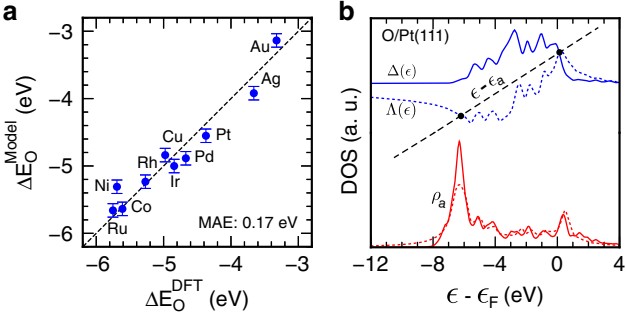

**Fig. 3 Model-predicted adsorption properties. a** DFT-calculated *O adsorption energies (atomic O as the reference) at transition-metal surfaces vs. model prediction using the posterior distribution of model parameters ($\vec{\theta}$, $\vec{\sigma}$). Error bars represent the standard deviation of model prediction with 1000 random draws from converged trajectories. **b** Projected density of states $\rho_a$ onto the $O_{2p}$ orbital from DFT calculations (solid) and model prediction (dashed) using the posterior means of model parameters, taking Pt(111) as an example. The graphical solution to the Newns–Anderson model is also shown, in which $\Delta(\epsilon)$ and $\Lambda(\epsilon)$ represent the chemisorption function, and $\epsilon_a$ is the adsorbate resonance energy level.

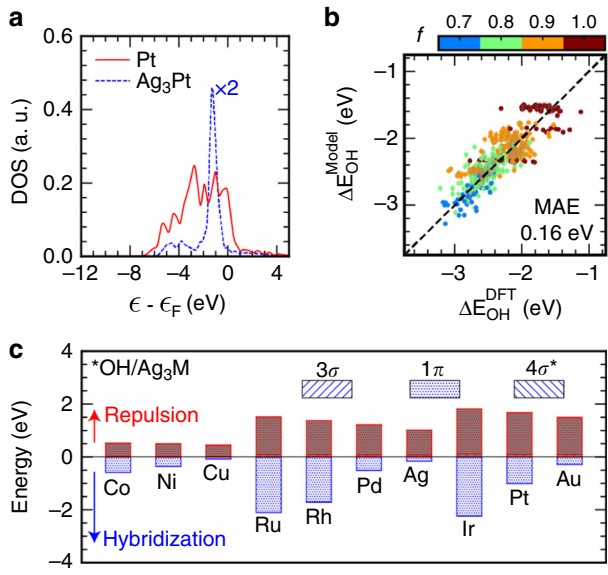

**Fig. 4 Model test and interpretation. a** The $d$-states of a transition-metal site exhibit diverse characteristics ranging from bulk-like semi-elliptic bands to free-atom-like discrete energy levels (Pt and Ag₃Pt as examples). **b** DFT-calculated vs. model-predicted adsorption energies of *OH at the atop site of {111}-terminated intermetallics and near-surface alloys with one standard deviation. **c** Partition of *OH adsorption energies at the M site of Ag₃M into orbital hybridization and orthogonalization of $3\sigma$, $1\pi$, and $4\sigma^*$ orbitals with the metal $d$-states. The $3\sigma$ and $4\sigma^*$ contributions are too small to be visible.

*OH prefers the nearly-parallel adsorption geometry on most of the $d$-metals to maximize the interaction of the $1\pi$ orbital with metal $d$-states, while *OH on Na(111) adsorbs more strongly in a up-straight orientation because of a lack of such directional interactions. This orbitalwise insight of chemical bonding could provide guidance in tailoring orbital-specific characteristics of the metal $d$-band for desired catalytic properties through site engineering. Despite an exclusive discussion about the $d$-metals, it is possible to extend the *Bayeschem* framework to $p$-block metals and alloys see Supplementary Fig. 11, unifying the reactivity theory of metal surfaces.

To conclude, we present the first Bayesian model of chemisorption by learning from ab initio adsorption properties. The model leverages the well-established $d$-band reactivity theory and a Newns–Anderson-type Hamiltonian for capturing essential physics of chemisorption processes. We demonstrated that the *Bayeschem* models of descriptor species, e.g., *O and *OH, optimized with pristine transition-metal data predicts adsorption energies at a diverse range of atomically-tailored metal sites with a MAE ~0.1–0.2 eV while providing uncertainty quantification. Incorporation of physics-based models into data-driven ML algorithms, e.g., deep learning, might hold the promise toward developing highly accurate while interpretable reactivity models. Furthermore, this conceptual framework can be broadly applied to unravel orbital-specific factors governing adsorbate–substrate interactions, paving the path toward design strategies to go beyond adsorption-energy scaling limitations in catalysis.

## Methods

**DFT calculations**. Spin-polarized DFT calculations were performed through Quantum ESPRESSO[39] with ultrasoft pseudopotentials. The exchange-correlation was approximated within the generalized gradient approximation (GGA) with Perdew–Burke–Ernzerhof (PBE)[40]. {111}-terminated metal surfaces were modeled using ($2 \times 2$) supercells with four layers and a vacuum of 15 Å between two images. The bottom two layers were fixed while the top two layers and adsorbates were allowed to relax until a force criteria of .1 eV/Å. A plane wave energy cutoff of 500 eV was used. A Monkhorst-Pack mesh of $6 \times 6 \times 1$ was used to sample the Brillouin zone, while for molecules and radicals only the Gamma point was used. Gas phase species of O and OH were used as the reference for adsorption energies of *O and *OH, respectively. The projected atomic and molecular density of states were obtained by projecting the eigenvectors of the full system at a denser $k$-point sampling ($12 \times 12 \times 1$) with a energy spacing 0.01 eV onto the ones of the part, as determined by gas-phase calculations. The convergence of DFT calculations was thoroughly tested to be within 0.05 eV. Further details and tabulated data can be found in the ref. [9].

**The $d$-band reactivity theory**. To revisit the $d$-band theory of chemisorption along with new developments, let's consider a metal substrate M in which electrons occupy a set of continuous states with one-electron wavefunctions $|k\rangle$ and eigen-energies $\epsilon_k$, and an isolated adsorbate species A with a valence electron described by an atomic wavefunction $|a\rangle$ at $\epsilon_a^0$, see Fig. 1. When the adsorbate is brought close to the substrate, the two sets of states will overlap and hybridize with each other. The strength of such interactions is determined by the coupling integral $V_{ak} = \langle a|\hat{\mathcal{H}}|k\rangle$, where $\hat{\mathcal{H}}$ is the system Hamiltonian. Within the Newns–Anderson model of chemisorption[17–19], $\hat{\mathcal{H}}$ is defined as,

$$\hat{\mathcal{H}} = \sum_\sigma \left\{ \epsilon_{a\sigma} n_{a\sigma} + \sum_k \epsilon_k n_{k\sigma} + \sum_k (V_{ak} c_{k\sigma}^\dagger c_{a\sigma} + H.c.) \right\}, \quad (1)$$

where $\sigma$ denotes the electron spin, $n$ is the orbital occupancy operator, and $c^\dagger$ and $c$ represent the creation and annihilation operator, respectively. The first two terms in Eq. (1) are the one-electron energies from the adsorbate and the substrate when they are infinitely separated in space. The last term captures the coupling, or intuitively electron hopping, between the adsorbate orbital $|a\rangle$ and a continuum of substrate states $|k\rangle$. If the one-electron states of the whole system can be described as a linear combination of the unperturbed adsorbate and substrate states, the one-electron Schrödinger equation can be solved using the Green's function approach[18]. In Fig. 1, we illustrate the chemisorption process of a simple adsorbate onto a $d$-block metal site characterized by delocalized $sp$-states and localized $d$-states[21]. The interaction of the adsorbate state at $\epsilon_a^0$ with the structureless $sp$-states, typically accompanied with electron transfer from/to the Fermi sea, results in a broadened resonance (or so-called renormalized adsorbate state) at an effective energy level $\epsilon_a$. Conceptually viewing chemical bonding as consecutive steps in Fig. 1, the renormalized adsorbate state then couples with the narrowly distributed $d$-states, shifting up in energies due to orbital orthogonalization that increases the kinetic energy of electrons and splitting into bonding and antibonding states. One important information from this framework is the evolving density of states projected onto the adsorbate orbital $|a\rangle$ upon adsorption

$$\rho_a(\epsilon) = \frac{1}{\pi} \frac{\Delta(\epsilon)}{[\epsilon - (\epsilon_a + \Lambda(\epsilon))]^2 + \Delta(\epsilon)^2}, \quad (2)$$

in which spin is neglected for simplicity. The effective adsorbate energy level, $\epsilon_a$, is determined by the image potential of a charged particle in front of conducting surfaces and the Coulomb repulsion between electrons in the same orbital[18]. The chemisorption function $\Delta(\epsilon)$ includes contributions from the $sp$-states and the $d$-

states

$$\Delta(\epsilon) = \pi \sum_k V_{ak}^2 \delta(\epsilon - \epsilon_k) = \Delta_0 + \Delta_d. \quad (3)$$

To simplify the matter, only the 2nd step interaction, i.e., the coupling of the renormalized adsorbate state with the substrate $d$-states, is explicitly considered in Eq. (2). As a new development in our approach, we include an energy-independent constant $\Delta_0$ along with $\Delta_d$ as the chemisorption function $\Delta(\epsilon)$. The inclusion of $\Delta_0$ provides a lifetime broadening of the adsorbate state, serving as a mathematical trick to avoid burdensome sampling of the resonance, i.e., the Lorentzian distribution $\tilde{\rho}_a$ from the 1st step interaction in Fig. 1. Accordingly, $\epsilon_a$ represents the renormalized adsorbate state. Attributed to the narrowness of a typical metal $d$-band, $\Delta_d$ can be simplified as the projected density of $d$-states onto the metal site $\rho_d(\epsilon)$ modulated by an effective coupling integral squared $V^2$, i.e., $\Delta_d \simeq \pi V^2 \rho_d(\epsilon)$. $\Lambda(\epsilon)$ is the Hilbert transform of $\Delta(\epsilon)$. In this framework, the interaction energy between the adsorbate and the substrate can be partitioned into two contributions, i.e., $\Delta E_0$ and $\Delta E_d$. $\Delta E_0$ is the energy change due to the interaction of the unperturbed adsorbate orbital(s) with the delocalized $sp$-states, while $\Delta E_d$ is the energy contribution from further interactions with the localized $d$-states of the substrate. Since all $d$-block metals have a similar, free-electron-like $sp$-band, $\Delta E_0$ can be approximated as a surface-independent constant albeit the largest contribution to bonding[21]. To calculate $\Delta E_d$, we include both the attractive orbital hybridization $\Delta E_d^{hyb}$ and repulsive orbital orthogonalization $\Delta E_d^{orth}$[29,41]:

$$\Delta E_d^{hyb} = \frac{2}{\pi} \int_{-\infty}^{\epsilon_F} \tan^{-1}\left[\frac{\Delta(\epsilon)}{\epsilon - \epsilon_a - \Lambda(\epsilon)}\right] d\epsilon - \frac{2}{\pi} \int_{-\infty}^{\epsilon_F} \tan^{-1}\left[\frac{\Delta_0(\epsilon)}{\epsilon - \epsilon_a}\right] d\epsilon$$
$$\Delta E_d^{orth} = 2(\langle \tilde{n}_a \rangle + f)\alpha\beta V_{ad}^2. \quad (4)$$

The constant 2 considers spin degeneracy of the orbital, $\langle \tilde{n}_a \rangle$ is the occupancy of the renormalized adsorbate state by integrating the Lorentzian distribution $\tilde{\rho}_a$ up to the Fermi level $\epsilon_F$ (taken as 0), and $f$ is the idealized $d$-band filling of the metal atom. The $\tan^{-1}$ is defined to lie between $-\pi$ to 0 since $\Delta_0$ is a nonzero constant across the energy scale $[-15, 15]$ eV. Thus there is no need to explicitly include localized states even if present below or above the $d$-band. In Eq. (4), $\alpha$ is termed the orbital overlap coefficient, i.e., $S \simeq \alpha|V|$, in which the overlap integral $S$ is linearly proportional to the coupling integral $V$ for a given orbital. Similarly, the effective coupling integral squared $V^2$ can be written as $\beta V_{ad}^2$, where $\beta$ denotes the orbital coupling coefficient and $V_{ad}^2$ characterizes the interorbital coupling strength when the bonding atoms are aligned along the $z$-axis at a given distance[42]. Its values of $d$-block metals relative to that of Cu are readily available on the Solid State Table[22]. It is important to note that $\beta$ is in the chemisorption function, which determines both the adsorption energy and adsorbate density of states, whereas $\alpha$ only affects the orbital orthogonalization energy since overlap was not explicitly considered.

**Bayesian learning**. Due to the computationally intensive nature of the MCMC algorithm, there is a need for a more efficient implementation of the Newns–Anderson model than what is obtained by Python and standard libraries like *SciPy* and *NumPy*. We make extensive use of Cython, a C++ extension to the standard Python, to speed up the performance (10–1000 times) of some CPU-intensive functions in the model, e.g., Hilbert transform. To perform MCMC sampling, we use *PyMC*, a flexible and extensible Python package which includes a wide selection of built-in statistical distributions and sampling algorithms[43], e.g., Metropolis-Hastings. A "burn-in" of the first half of the samplings and then thinning (1 out of 5 samplings) was performed to ensure that subsequent ones are representative of the posterior distribution. Convergence of our MCMC-based sampling was verified using parallel chains[28]. The MCMC sampling results can be directly visualized using *corner*, a open-source Python module. We took *Normal* for floating-point variables unrestricted in sign, *LogNormal* for non-negative parameters, and *Uniform* for others. $\Delta E_0$ and $\epsilon_a$ can be estimated from DFT calculations of the adsorbate on a simple metal, e.g., sodium (Na) at the face-centered cubic (fcc) phase. Specifically, for *O, we used $\Delta E_0 \sim N(-5.0, 1)$, $\epsilon_a \sim N(-5, 1)$, $\Delta_0 \sim LN(1, 0.25)$, $\beta \sim LN(2, 1)$, and $\alpha \sim U(0, 1)$. For *OH, we used $\Delta E_0 \sim N(-3.0, 1)$, $\epsilon_a^{3\sigma} \sim N(-6, 1)$, $\epsilon_a^{1\pi} \sim N(-2, 1)$, and $\epsilon_a^{4\sigma^*} \sim N(4, 1)$. We assume that the predicted adsorption properties from Eqs. (2) and (4) are subject to independent normal errors. Specifically, for the property $Y$ and the surface $i$ we have

$$Y_i = \hat{Y}_i(\vec{\theta}) + \sigma\epsilon_i, \ i = 1, 2, \ldots, n, \quad (5)$$

where $\epsilon_i$ is an independent and standard normal random variable and $\sigma$ is the standard deviation, allowing for a mismatch between the model prediction $\hat{Y}_i(\vec{\theta})$ and the DFT ground truth $Y_i$. In this approach, we define the likelihood function of the property $Y$ from $n$ observations[44]

$$P(Y|\vec{\theta}, \sigma) \propto \sigma^{-n} \exp\left[-\frac{1}{2\sigma^2}\sum_{i=1}^{n}\left\{Y_i - \hat{Y}_i(\vec{\theta})\right\}^2\right], \quad (6)$$

where the sum runs over $n$ training samples for the property $Y$, which is either the projected density of states onto an adsorbate orbital or adsorption energies. For adsorption energies, $Y_i$ and $\hat{Y}$ are scalar values with no ambiguity. For projected density of states, it is a vector of paired values, i.e., the one-electron energy of a state

and its probability density, thus deserving a clarification. The mean squared residuals of model prediction from Eq. (2) for the surface $i$ is used as $\{Y_i - \hat{Y}_i(\vec{\theta})\}^2$ in Eq. (6). To compute the transition probability of each MCMC step, we define the sum of the (negative) logarithm of the likelihood functions corresponding to projected density of states onto each adsorbate orbital and binding energies with a hyper parameter $\lambda$ adjusting the weight of two contributing metrics, i.e., $-\ln(P_{\Delta E}) - \lambda \sum \ln(P_{\rho_a})$. To optimize this parameter, we varied it on a grid of 1.0e−3, 1.0e−2, 1.0e−1, and 1, and found that 1.0e−2 is the optimal value to obtain the best performance in adsorption energy prediction.

## Data availability

The training data of metal surfaces used for model development is available at the Github repository https://github.com/hlxin/bayeschem while the test data are from the article https://doi.org/10.1039/C7TA01812F10.1039/C7TA01812F.

## Code availability

The complete code of *Bayeschem* is available at a Github repository https://github.com/hlxin/bayeschem for public access.

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

## Acknowledgements

S.W., H.S.P., and H.X. acknowledge the financial support from the NSF CAREER program (CBET-1845531). The computational resource used in this work is provided by the advanced research computing at Virginia Polytechnic Institute and State University. H.X. acknowledges the insightful discussion with Prof. John Kitchin from Carnegie Mellon University that inspired the work.

## Author contributions

S.W. and H.S.P. equally contributed to the work. H.X. supervised the research. S.W. and H.X. conceived the idea and designed the general approach. S.W. and H.S.P. conducted DFT calculations and coding. S.W. and H.S.P. performed the detailed analysis. All authors revised the manuscript.

## Competing interests

The authors declare no competing interests.
