## [Peer Review File · Nature Communications]

Reviewers' comments:

Reviewer #1 (Remarks to the Author):

NCOMMS-20-29836

Wang, Yillai and Xin present a Bayesian inference approach to estimate the parameters of the d-band adsorption model for metals, instead of predicting adsorption/bonding energies as commonly done in literature. I found the approach quite refreshing, as the way the parameters are predicted aids the interpretability of the results, something often ignored in recent literature about machine-learning in heterogeneous catalysis. They also escaped the traditional adsorption-energy scaling relationships, which are still the most interpretable and accurate models available.

Despite the advantages, my general impression is that the Bayesian inference approach presented here is quite limited in scope and a bit too convoluted, missing significant gains in accuracy or interpretability as detailed below:

First, the authors report a MAE of 0.16 eV for extrapolating the adsorption energy of OH from O. Taking the simplest model possible, which is a linear regression between the energies of OH and O (Abild-Pedersen et al., Phys. Rev. Lett. 2007), applying it to in-house local data, I got a MAE of 0.17 eV. That model is far more interpretable in terms of number of bonds being broken and could be applied and validated in less than 5 minutes. I still recognize that the authors did outperform black-boxed data-driven ML models in terms of accuracy and interpretability.

Second, it is suboptimal that the interpretability depends on the d-band model. Many promising metals or alloys for "metal-catalyzed electrochemical O₂ reduction, CO₂ reduction, H₂ oxidation in alkaline electrolytes, etc.", literally the applications listed by the authors, have Cu, Ag, and Au (in which the d-band model starts to shake), p-block metals, or f-block metals.

Third, the molecules used in the model, O vs OH, are too simple to be used as benchmark for heterogeneous catalysis. Thus it is very hard to contrast its accuracy with other pre-existing models.

Also to consider:

The manuscript does not mention anything related to the conformation of the adsorbates. In some metals, O adsorbs preferentially on fcc sites, while in others it prefers hcp sites. OH may adsorb on fcc, hcp, bridge, and top sites, and each of these sites may be the preferred one for a given metal or alloy. How their model deals with this issue? Because the local coordination of O and (specially) OH may have a strong influence on the ΔE_0 parameter and the orbitalwise contributions analyzed by the authors. ΔE_0 is rather constant among metals, but that is only true if the adsorbate ground state is the same on all metals.

The literature review has a lot of room for improvement. There are very few recent references: only 4 from 2018, 3 from 2019 (one of them, #31, is in ChemRxiv), and none from 2020. Many lacked a doi or information about volume and pages, giving the impression that it was done two years ago and then forgotten. General, interpretable, and high-accuracy ML models are a hot topic nowadays. I can think of at least 9 articles in the last 12 months developing "interpretable" approaches, each of them from a different group in 6 different countries. Yet, none of them are mentioned or benchmarked.

The DFT part does not look well converged. First, the k-point sample is probably not converged for some of the metals used in the analysis (like Cu and perhaps Co, Ni, Rh, and Ru) for which the authors would need at least 8x8 or 9x9 kpoints for the size of their cell during the relaxation. However, the kpoint grid used for the projected density of states is OK. Second, the force criterion

of 0.1 eV/Å is also too broad. Being their adsorbates so simple, they can attain much tighter accuracy (around 0.02 or even 0.01 eV/Å) without considerable computational cost (maybe 2-5 additional ionic steps). Third, a 2×2 cell is too small for such "strong" adsorbates like O and OH. At that surface concentration the adsorbates interact with their periodic images.

"Bayesian learning"?

This sentence is very difficult to understand: "Compared with a full quantum-mechanics treatment of many-body systems, the simplicity of physics-inspired descriptors comes at a cost of limited generalization, particularly for high-throughput materials screening in which variations of site composition and configuration are sufficiently large to invalidate the perturbation approximation."

Reviewer #2 (Remarks to the Author):

Wang et al. develop a Physics-based surrogate model for computing the binding energy (and other adsorption properties) of small adsorbates on transition metal catalysts. The authors specifically employ the d-band model (with the Newns-Anderson-Grimley chemisorption model) which they suitably modify so that the chemisorption system can be defined on the basis of five parameters that are then be fit to DFT-calculated properties (binding energy, density of states). To learn the parameters, the authors adopt a Bayesian learning formalism. The authors show that this method allows for learning simplified chemisorption models for small adsorbates which can then be applied to compute the binding energy on a large number of alloy surfaces. The idea, in general, is quite intriguing as incorporating physics into a data-driven model allows for learning from small number of data sets, which in itself can be quite transformative and hence this work would be of interest to the general readership of this journal. However, some questions remain, as discussed below, that need to be addressed before this work can be published.

1. An important question is the transferability of this model to different sites and species which has not been adequately discussed. It appeared that models were created for an individual species at a specific site. Is this true? For instance, how well does this model work to calculate the binding energy on a bridge site (hcp site) from training only on say fcc sites? This aspect was not clearly discussed. It is perhaps OK for a data-driven model to not allow for this, however, more sophisticated deep learning approaches, such as graph convolutional networks, in principle provide this resolution (although also need more data). Relatedly, should models be trained for each intermediate independently, or is there a more general formalism that allows to build a single model for multiple intermediates? Furthermore, can a model formalism be developed so different surface facets can be simultaneously captured within a single model? Finally, how is this formalism expected to work for other (larger) species? Discussion of these aspects would add a clearer perspective of the potential of this work.

2. In the extension of the OH model to alloys, how is V_{ad} computed? This seems to be taken from a table for monometallic catalyst, therefore its calculation for alloys is unclear.

3. It appears from the formalism that the orbital orthogonalization energy is directly related to the product of α and square root of β , both of which are parameters to be learned. In such a case, estimating these parameters individually seems unnecessary as they are likely heavily correlated.

4. The authors do not seem to be using the power of Bayesian approach to the fullest extent. Rather than just reporting mean values, they should report 95% confidence regions of their predictions in Figures 3b, 4a-c to understand the reliability of the predicted properties.

Reviewer #3 (Remarks to the Author):

The authors describe a novel approach to predicting adsorption energies of molecules on metal surfaces. They demonstrate that by integrating data-driven methods (Bayesian learning) along with physics-based models, they obtain predictive models that are also physically insightful because they are learning the values of inputs to physical models. The paper is well-written and should be impactful for the computational design of catalysts and generally in the space of machine learning for physical science and engineering applications. I believe the paper is suitable for Nature Communications after minor revisions.

Generally, I recommend the authors make the language a bit more accessible to a general audience. The paper reads clearly to a theoretical chemist, but I fear that others may miss some of the main points of the paper because there is quite a bit of derivation involved. This could potentially be addressed by simply expanding the length of certain sections to make clear what are the main points of each section in accessible terms.

In a similar spirit, the notation does get a bit unwieldy in places. For example, the d-band reactivity theory section does not define certain parameters (e.g., ϵ_a). Most of the figures also have undefined notation – θ_0 in Figure 2b, several quantities in Figure 3b, “f” in Figure 4b, etc. These should be explicitly defined within the respective figure captions.

Reviewer # 1

Comments: Wang, Pillai and Xin present a Bayesian inference approach to estimate the parameters of the d-band adsorption model for metals, instead of predicting adsorption/bonding energies as commonly done in literature. I found the approach quite refreshing, as the way the parameters are predicted aids the interpretability of the results, something often ignored in recent literature about machine-learning in heterogeneous catalysis. They also escaped the traditional adsorption-energy scaling relationships, which are still the most interpretable and accurate models available.

Response: We thank the reviewer for positive comments. Indeed, the approach goes beyond the purely regression-based machine learning methods that are popularized in recent years by many groups including ourselves. Currently, there is a lot of interest in the field to make machine learning models more interpretable. As mentioned by the reviewer, scaling relations are widely used in heterogeneous catalysis. It is insightful in terms of the number of valence electrons that participate in chemical bonding. However, it does not provide a fundamental connection between the electronic structure of an adsorption site and the bonding strength of a given adsorbate. Our approach fills the gaps and provides orbitalwise insights into chemical bonding at metal surfaces that could be eventually leveraged to break scaling relations.

Despite the advantages, my general impression is that the Bayesian inference approach presented here is quite limited in scope and a bit too convoluted, missing significant gains in accuracy or interpretability as detailed below:

Response: We thank the reviewer for allowing us to discuss further details of the approach. In the revised manuscript, we made significant efforts to broaden its scope to different adsorption sites and complex adsorbates, as shown in the point-by-point response. We have an ongoing effort in the code development to make the *Bayeschem* an ase-compatible package (<https://wiki.fysik.dtu.dk/ase/ase/atoms.html>) that can be easily used for training physical models and interpreting results, see the Github repository <https://github.com/hlxin/bayeschem>. In terms of the accuracy, we showed a prediction error of *OH adsorption energies on alloy surfaces comparable to linear regression and previous nonlinear machine learning models. A significant gain of the approach is the physical interpretability. Compared to purely data-driven machine learning models, the *Bayeschem* allows us to understand the effect of electronic structure on chemical bonding at metal surfaces and deconvolute orbitalwise contributions to the total adsorption energy. We emphasize this point in the following response.

Q1: First, the authors report a MAE of 0.16 eV for extrapolating the adsorption energy of OH from O. Taking the simplest model possible, which is a linear regression between the energies of OH and O (Abild-Pedersen et al., Phys. Rev. Lett. 2007), applying it to in-house local data, I got a MAE of 0.17 eV. That model is far more interpretable in terms of number of bonds being

broken and could be applied and validated in less than 5 minutes. I still recognize that the authors did outperform black-boxed data-driven ML models in terms of accuracy and interpretability.

Response: We thank the reviewer for bringing up the linear adsorption-energy scaling relations in prediction of adsorption energies. There is a confusion that needs to be clarified. For the *OH model shown in the manuscript, there was no extrapolation from *O. We did not use the *O adsorption energy or the *O adsorption model to predict *OH adsorption energies. The linear adsorption-energy scaling between *O and *OH holds very well on pure metals with R^2 : 0.95. However, there is a significant deviation for transition-metal alloys, see H. Xin, S. Linic, *J. Chem. Phys.* **132**, 221101–221101–4 (2010). In our model scheme, an individual set of parameters is obtained for each adsorbate at a given site. Our goal is to develop the connection between the electronic structure of a surface site and the adsorption energy, whereas the scaling relations connect the adsorption energies of different adsorbates (*O and *OH in this case). With the *Bayeschem* model, we are able to resolve orbitalwise contributions to the total adsorption energy and also quantify the importance of Pauli repulsion and orbital hybridization. Probing those finer details thoroughly is not possible with scaling relations and any other previous models. Unraveling origin of adsorbate-substrate interactions will eventually lead to better understanding of the scaling relations at the electronic structure level, and devise strategies to go beyond the scaling relations that are posing limitations of attainable catalytic performance.

Changes: We add the following into the main text:

“In this model scheme, an individual set of parameters is obtained for the adsorbate at a given site. Compared to the linear adsorption-energy scaling relations that link adsorption energies of different adsorbates, *Bayeschem* creates the connection between the electronic structure of a surface site and the adsorption energy.”

Q2: Second, it is suboptimal that the interpretability depends on the d-band model. Many promising metals or alloys for "metal-catalyzed electrochemical O₂ reduction, CO₂ reduction, H₂ oxidation in alkaline electrolytes, etc.", literally the applications listed by the authors, have Cu, Ag, and Au (in which the d-band model starts to shake), p-block metals, or f-block metals.

Response: The reviewer is right about the limitations of the d-band model for coinage metals. From a traditional view, the d-band model emphasizes the importance of one or two physical factors, such as the d-band center. It is mainly used for understanding reactivity trends of metal surfaces or predicting new catalysts using DFT-calculated d-band centers. We built on the basic idea of the d-band reactivity theory in interpreting the sp-band and d-band interactions with an adsorbate. Nevertheless, we employed the full Newns-Anderson-Grimley (NAG) model in the formulation of adsorbate density of states and adsorption energies. The physical interpretability offered by the NAG model goes well beyond the linear relationships with the d-band center or other electronic descriptors, as demonstrated in Fig. 4. Its extension to p-block or f-block metals has yet to be developed. However, we found that there is a linear relationship of the p-band center of doped Bi-surfaces with the adsorption energies of *CHO intermediate, one of the key intermediates in the context of CO₂ electroreduction. We believe that it is possible to extend the NAG model with the *Bayeschem* to p-block metals and alloys, unifying the reactivity theory of metal surfaces.

Changes: We add the following into the conclusions: “Despite an exclusive discussion about the d-metals, it is possible to extend the *Bayeschem* framework to p-block metals and alloys,

unifying the reactivity theory of metal surfaces.” We also added the following figure to the SI for reviewers only.

Figure S1. The adsorption energy of $^*\text{CHO}$ at the atop Bi site of M-doped Bi(001) surfaces vs. the Bi p -band center. The inset shows model structures of M-doped Bi(001).

Q3: The molecules used in the model, O vs OH, are too simple to be used as benchmark for heterogeneous catalysis. Thus it is very hard to contrast its accuracy with other pre-existing models.

Response: We thank the reviewer for pointing out the simplicity of those adsorbates used in the model development. We chose them purposely to demonstrate the approach. It is also important to emphasize that they are widely used reactivity descriptor species in heterogeneous catalysis. As elaborated in the manuscript, the approach can be easily extended to other adsorbates. To further demonstrate this, we developed the Bayesian chemisorption model for $^*\text{OOH}$ at metal surfaces.

Changes: We added the following into the main text: “The approach can be easily extended to more complex adsorbates than $^*\text{O}$ and $^*\text{OH}$, e.g., $^*\text{OOH}$, without losing its generalizability in the development workflow.” We are providing the model details in the SI for reviewers since it is in another effort specifically investigating the origin of the scaling relation between $^*\text{OH}$ and $^*\text{OOH}$ adsorption energies.

Q4: The manuscript does not mention anything related to the conformation of the adsorbates. In some metals, O adsorbs preferentially on fcc sites, while in others it prefers hcp sites. OH may adsorb on fcc, hcp, bridge, and top sites, and each of these sites may be the preferred one for a given metal or alloy. How their model deals with this issue? Because the local coordination of O and (specially) OH may have a strong influence on the ΔE_0 parameter and the orbitalwise contributions analyzed by the authors. ΔE_0 is rather constant among metals, but that is only true if the adsorbate ground state is the same on all metals.

Response: We thank the reviewer for pointing out the site preference complexities in surface chemistry. Within the current approach, one set of parameters, i.e., one model, is obtained for the adsorbate at a particular adsorption site. It is a choice for simplicity since optimizing model

parameters is rather straightforward once a small training dataset is in place. The reviewer is right about the dependence of model parameters, e.g. ΔE_0 , on the adsorption configuration. To show this we have optimized the *Bayeschem* model of *O at the atop configuration, see Fig. S5-S7.

Changes: We added the following into the main text: “To demonstrate the robustness and generalizability of the approach, we have also optimized the *Bayeschem* model of *O at the atop configuration, see Fig. S5-S7.

Q5: The literature review has a lot of room for improvement. here are very few recent references: only 4 from 2018, 3 from 2019 (one of them, #31, is in ChemrXiv), and none from 2020. Many lacked a doi or information about volume and pages, giving the impression that it was done two years ago and then forgotten. General, interpretable, and high-accuracy ML models are a hot topic nowadays. I can think of at least 9 articles in the last 12 months developing "interpretable" approaches, each of them from a different group in 6 different countries. Yet, none of them are mentioned or benchmarked.

Response: Thanks the reviewer for the suggestion. There are lots of interest in developing general, interpretable, and high-accuracy ML models in recent years. From our best knowledge, there are two types. One is employing physical factors in regression learning, e.g., electronic structure of an adsorption site (M. M. Montemore, et al., *Catal. Sci. Technol.* **10**, 4467–4476, 2020) or coordination numbers (K. Tran, et al., *Nature Catalysis.* **1**, 696–703, 2018). The ML model developed with this type of features is more interpretable than others using graph features or atomic properties. Another type is using learning algorithms, e.g., compressed sensing (M. Andersen, et al., *ACS Catal.* **9**, 2752–2759, 2019) or convolutional neural networks (S. Back, et al., *J. Phys. Chem. Lett.* **10**, 4401–4408, 2019; G. H. Gu, et al., *J. Phys. Chem. Lett.* **11**, 3185–3191, 2020), to extract high-level features that can be used for model interpretation. We have expended references in the revised manuscript. From our best knowledge, there is no interpretable ML model that is built on the well-established theory of surface chemisorption while leveraging machine learning techniques for model optimization. We also fixed those missing volumes, which are due to outdated paperpile references.

Q6: The DFT part does not look well converged. First, the k-point sample is probably not converged for some of the metals used in the analysis (like Cu and perhaps Co, Ni, Rh, and Ru) for which the authors would need at least 8x8 or 9x9 kpoints for the size of their cell during the relaxation. However, the kpoint grid used for the projected density of states is OK. Second, the force criterion of 0.1 eV/Å is also too broad. Being their adsorbates so simple, they can attain much tighter accuracy (around 0.02 or even 0.01 eV/Å) without considerable computational cost (maybe 2-5 additional ionic steps). Third, a 2x2 cell is too small for such "strong" adsorbates like O and OH. At that surface concentration the adsorbates interact with their periodic images.

Response: We have performed further convergence tests of k-points, force criteria, and unit cell size and conclude that the DFT calculations are converged within 0.05 eV. For example, the adsorption energy of *OH at Cu(111) using 8x8x1 k-points is .03 eV weaker than that from the 6x6x1 k-points. Increasing the unit cell size from 2x2x4 to 3x3x4 Cu(111) only changes the *OH adsorption energy by -0.05 eV. Changing the force convergence from 0.1 eV/Å to 0.03 eV/Å changes the *OH adsorption energy on Cu(111) by .03 eV.

Q7: "Bayesian learning"?

Response: We have fixed the typo.

Q8: This sentence is very difficult to understand: "Compared with a full quantum-mechanics treatment of many-body systems, the simplicity of physics-inspired descriptors comes at a cost of limited generalization, particularly for high-throughput materials screening in which variations of site composition and configuration are sufficiently large to invalidate the perturbation approximation."

Changes: We have changed "Compared with a full quantum-mechanics treatment of many-body systems, the simplicity of physics-inspired descriptors comes at a cost of limited generalization, particularly for high-throughput materials screening in which variations of site composition and configuration are sufficiently large to invalidate the perturbation approximation." to "Compared with a full quantum-mechanics treatment of many-body systems, the simplicity of physics-inspired descriptors comes at a cost of limited generalization, particularly for high-throughput materials screening."

Reviewer # 2

Comments: Wang et al. develop a Physics-based surrogate model for computing the binding energy (and other adsorption properties) of small adsorbates on transition metal catalysts. The authors specifically employ the d-band model (with the Newns-Anderson-Grimley chemisorption model) which they suitably modify so that the chemisorption system can be defined on the basis of five parameters that are then be fit to DFT-calculated properties (binding energy, density of states). To learn the parameters, the authors adopt a Bayesian learning formalism. The authors show that this method allows for learning simplified chemisorption models for small adsorbates which can then be applied to compute the binding energy on a large number of alloy surfaces. The idea, in general, is quite intriguing as incorporating physics into a data-driven model allows for learning from small number of data sets, which in itself can be quite transformative and hence this work would be of interest to the general readership of this journal. However, some questions remain, as discussed below, that need to addressed before this work can be published.

Response: We thank the reviewer for positive comments of the work. We have addressed the raised questions below point-by-point.

Q1: An important question is the transferability of this model to different sites and species which has not been adequately discussed. Is this true? For instance, how well does this model work to calculate the binding energy on a bridge site (hcp site) from training only on say fcc sites? Relatedly, should models be trained for each intermediate independently, or is there a more general formalism that allows to build a single model for multiple intermediates? Furthermore, can a model formalism be developed so different surface facets can be simultaneously captured within a single model? Finally, how is this formalism expected to work for other (larger) species? Discussion of these aspects would add a clearer perspective of the potential of this work.

Response: The reviewer brought up a very important question regarding the transferability of the approach. In regression-based machine learning, it is possible to have a unified model by designing the learning scheme and feeding more data that contain multiple sites, facets, and

adsorbates. To have a *Bayeschem* model that can transfer from one site to another, it requires training data of different sites as well. Besides that, two developments are needed. 1) We need to employ the site-specific electronic structure that captures the local environment of the binding site rather than the ensemble average of site atoms. This can be done by using the group orbital density of states concept (Alford, R.; Kelly, K.; Boore, D. M. Accuracy of finite-difference modeling of the acoustic wave equation. Geophysics 1974, 39, 834). 2) Another one that is more challenging to resolve is the linking of model parameters from different sites within a unified framework. At the moment, one set of parameters, i.e., one model, is optimized per adsorbate on a given adsorption site, e.g., atop, fcc/hcp hollow. It should be noted that models can be quickly built in this fashion. As an example, we have optimized the model of *O at the atop adsorption site, see Fig. S5-S7.

While a unified *Bayeschem* model for multiple intermediates would be insightful, the exact approach to acquire that complexity is not known. One possible solution would be to use the scaling relations since these would connect different intermediates. However, the model would be highly convoluted and the model parameters are not intrinsic to a particular adsorbate and adsorption configuration, prohibiting its physical interpretation.

The reviewer also mentioned whether it is possible to have one model for different surface facets. It burns down to a similar task for different sites as we discussed. The group orbital density of d-states should be used as the site electronic structure. And the model parameters will be designed to be aware of the local coordination environment. Again, it is a direction worthy of exploring for further understanding of chemical bonding. The basic framework and the tools developed here will allow researchers to explore the ideas in those directions.

To show the model's applicability for larger molecules we have also built the model for *OOH and see that the model captures adsorption energy and molecular orbital DOS very well without losing the generalizability of the approach. The results are provided in the SI for reviewers.

Changes: We provide the following changes in the SI for publications.

Figure S5. The co-variance of the joint posterior distribution for each parameter pair and 1D histogram of single-orbital model parameters for oxygen adsorbed on atop (ΔE_0 ,

ϵ_a , Δ_0 , α , and β) from MCMC simulations for *O adsorption at the atop site of the close-packed, pristine transition-metal surfaces.

Figure S6. DFT-calculated *O adsorption energies at the atop site of metal surfaces vs. model prediction using the posterior distribution of single-orbital model parameters (θ , σ). Error bars represent the standard deviation of prediction based on Bayesian sampling.

Figure S7. Symmetry resolved projected O_{2p} density of states from DFT calculations (solid) and model prediction (dashed) for the atop site using the Bayesian-learned posterior means of model parameters. Metal d -states are also shown.

Q2: In the extension of the OH model to alloys, how is V_{ad} computed? This seems to be taken from a table for monometallic catalyst, therefore its calculation for alloys is unclear.

Response: The coupling matrix element V_{ad} relative to Cu is directly from the solid state table in the Harrison book. This is an approximation that can be relaxed by incorporating distance dependence into the model. We have further clarified this in the revised manuscript.

Changes: We have added the following additional information to the main text to clarify this point : “The coupling matrix element V_{ad} for alloys is assumed to be constant from the Solid State Table. Its dependence on the local chemical environment can be incorporated into the model using the tight-binding approximation. ”

Q3: It appears from the formalism that the orbital orthogonalization energy is directly related to the product of alpha and square root of beta, both of which are parameters to be learned. In such a case, estimating these parameters individually seems unnecessary as they are likely heavily correlated.

Response: Actually, those two parameters alpha and beta are not correlated. Beta is in the chemisorption function which determines both the adsorption energy and adsorbate density of states. Whereas alpha only affects the orbital orthogonalization energy. It can also be seen from the histograms of posterior distributions that beta and alpha for both the *OH and *O models are not correlated.

Changes: We have added the following to emphasize the point raised by the reviewer. “It is important to note that Beta is in the chemisorption function which determines both the adsorption energy and adsorbate density of states, whereas alpha only affects the orbital orthogonalization energy.”

Q4: The authors do not seem to be using the power of Bayesian approach to the fullest extent. Rather than just reporting mean values, they should report 95% confidence regions of their predictions in Figures 3b, 4a-c to understand the reliability of the predicted properties.

Response: This is an important point. We have updated the Fig. 4(b) so the standard deviation of predicted *OH adsorption energies is shown. Fig. 3(b) is not influenced because the uncertainty of those model parameters are too small to be visualized.

Changes: The following text has also been added: “The standard deviation of predicted *OH adsorption energies from the posterior distribution of model parameters is marked for uncertainty quantification.”

Reviewer # 3

Comments: The authors describe a novel approach to predicting adsorption energies of molecules on metal surfaces. They demonstrate that by integrating data-driven methods (Bayesian learning) along with physics-based models, they obtain predictive models that are also physically insightful because they are learning the values of inputs to physical models. The paper is well-written and should be impactful for the computational design of catalysts and generally in the space of machine learning for physical science and engineering applications. I believe the paper is suitable for Nature Communications after minor revisions.

Response: We are grateful for the reviewer’s positive comments on the application of the *Bayeschem* approach in catalyst design and its broader impact in the physical science and engineering.

Q1: Generally, I recommend the authors make the language a bit more accessible to a general audience. The paper reads clearly to a theoretical chemist, but I fear that others may miss some

of the main points of the paper because there is quite a bit of derivation involved. This could potentially be addressed by simply expanding the length of certain sections to make clear what are the main points of each section in accessible terms.

Response: We thank the reviewer for constructive suggestions to make the work more accessible to a general audience.

Changes: We have revised the main text with more accessible terms and also expanded the Method section to make the derivation easier to follow. We have changed “In Bayes' view, those parameters are not deterministic point values, but rather a probabilistic distribution reflecting not necessarily the random nature of physical variables but rather the uncertainty.” to “In Bayes' view, those parameters are not deterministic point values, but rather probabilistic distributions reflecting the uncertainty of physical variables.”

Q2: In a similar spirit, the notation does get a bit unwieldy in places. For example, the d-band reactivity theory section does not define certain parameters (e.g., ϵ^0_a). Most of the figures also have undefined notation – θ_0 in Figure 2b, several quantities in Figure 3b, “f” in Figure 4b, etc. These should be explicitly defined within the respective figure captions.

Response: We thank the reviewer for pointing out the notation problem. Those symbols were originally defined in the main text. However, it was later moved to the end of the manuscript in compliance with the journal format. In the revised manuscript, we have fixed this issue by defining it when it first appears in the main text.

Changes: In Fig. 2b, we defined $\vec{\theta}_0$ in the caption by adding “ $\vec{\theta}_0$ represents the initial guess of model parameters.” We also changed the notations correspondingly in the main text.

REVIEWERS' COMMENTS

Reviewer #1 (Remarks to the Author):

Most of my previous comments were addressed. I have only two minor revisions:

From authors reply to Q2: "...We believe that it is possible to extend the NAG model with the Bayeschem to p-block metals and alloys, unifying the reactivity theory of metal surfaces". As a first minor revision, the authors need to include their results on Bi-doped alloys in their Supplementary Information for publication (not for Reviewers only). This way, any future reader will see that such claim is based on evidence rather than beliefs, and that the Bayeschem model is more "general" than others on the literature.

The tests on *OOH do not provide any new information and can be kept as "only for Reviewers".

Q6: The accuracy test is poorly done. For instance, *OH was selected instead of *O, where the later is expected to have a much stronger interaction with its periodic image, and therefore error bars higher than 0.1 eV. It is also unnecessary to introduce uncertainty with a loose criterion when converging by forces, specially with such simple adsorbates. Yet, the general conclusions of this article will not change by refining the DFT data; only the error bars may get lower and 0.16 eV is already a good mark. Thus, as the second minor revision, please report instead the accuracy test with *O instead of *OH.

Reviewer #2 (Remarks to the Author):

The authors have adequately addressed the questions raised by this reviewer and their work can be accepted for publication. The reviewer reiterates the value of building a physics-based ML model for multiple intermediates and adsorption sites; this work is a step in the right direction.

Reviewer #1:

Comments: Most of my previous comments were addressed. I have only two minor revisions:

From authors reply to Q2: "...We believe that it is possible to extend the NAG model with the Bayeschem to p-block metals and alloys, unifying the reactivity theory of metal surfaces".

As a first minor revision, the authors need to include their results on Bi-doped alloys in their Supplementary Information for publication (not for Reviewers only). This way, any future reader will see that such claim is based on evidence rather than beliefs, and that the Bayeschem model is more "general" than others on the literature.

The tests on *OOH do not provide any new information and can be kept as "only for Reviewers".

Response: We thank the reviewer for this suggestion. We have now moved this Figure to the SI.

Changes: We add the following into the SI as Figure S11:

Figure S12. The adsorption energy of *CHO at the atop Bi site of M-doped Bi(001) surfaces vs. the Bi p-band center of all states. The inset shows model structures of M-doped Bi(001).

Q6: The accuracy test is poorly done. For instance, *OH was selected instead of *O, where the later is expected to have a much stronger interaction with its periodic image, and therefore error bars higher than 0.1 eV. It is also unnecessary to introduce uncertainty with a loose criterion when converging by forces, specially with such simple

adsorbates. Yet, the general conclusions of this article will not change by refining the DFT data; only the error bars may get lower and 0.16 eV is already a good mark. Thus, as the second minor revision, please report instead the accuracy test with *O instead of *OH.

Response: We have performed further convergence tests of k-points, force criteria, and unit cell size for *O species. Similar to *OH, the DFT calculated *O adsorption energies are converged within 0.1 eV. For example, the adsorption energy of *O at Cu(111) using 8x8x1 k-points is .06 eV stronger than that from the 6x6x1 k-points. Increasing the unit cell size from 2x2x4 to 3x3x4 Cu(111) only changes the *O adsorption energy by -0.07eV. Changing the force convergence from 0.1 eV/Å to 0.03 eV/Å changes the *O adsorption energy on Cu(111) by .001 eV.